# Development of an Anomaly Alert System Triggered by Unusual Behaviors at Home

**DOI:** 10.3390/s21165454

**Published:** 2021-08-12

**Authors:** Roxana Rodriguez-Goncalves, Angel Garcia-Crespo, Carlos Matheus-Chacin, Adrian Ruiz-Arroyo

**Affiliations:** Innovation Promotion and Technological Development Institute, University Carlos III of Madrid, 28911 Leganes, Spain; roxrodri@inf.uc3m.es (R.R.-G.); carlosalberto.matheus@uc3m.es (C.M.-C.); adruiza@inst.uc3m.es (A.R.-A.)

**Keywords:** Internet of Things, elderly, expert system, homecare, ambient sensors

## Abstract

In many countries, the number of elderly people has grown due to the increase in the life expectancy of the population, many of whom currently live alone and are prone to having accidents that they cannot report, especially if they are immobilized. For this reason, we have developed a non-intrusive IoT device, which, through multiple integrated sensors, collects information on habitual user behavior patterns and uses it to generate unusual behavior rules. These rules are used by our SecurHome system to send alert messages to the dependent person’s family members or caregivers if their behavior changes abruptly over the course of their daily life. This document describes in detail the design and development of the SecurHome system.

## 1. Introduction

Currently, in many countries, the number of elderly people is growing due to the increase in the life expectancy of the population. Figure 1 shows the latest World Bank update on life expectancy in some countries and at the level global (as of 9 April 2020). For example, in Spain, life expectancy at birth was 81.18 in 2008, while by 2018, it had increased to 83.33 (a difference of 2.15 over 10 years) [1].

This life expectancy is higher for women than men and is one of the reasons (in addition to divorce and personal preferences for living alone) that, in developed countries, the number of one-person homes for the elderly is constantly growing. In Spain, according to the data issued in July 2019 by the National Statistics Institute (INE), a total of 4,829,600 people were living alone (10.2% of the population), of whom 41.6% were people over 65 (with 72.3% of them being women) [2]. INE projections indicate that in 15 years, the number of people living alone will increase by 18.3% [3]. This is also the case at the European level, with an increasing number of older people in the EU living alone (especially older women). In 2020, it was indicated that 40.2% of women over 65 live alone, while 21.8% of men over 65 live alone [4].

Many studies have shown that living alone can have negative consequences on elderly people’s quality of life. Analysis of some cross-sectional studies has indicated that living alone is a risk factor for frailty in older people [5]. In addition, some studies indicate that living alone is a risk factor for depression. The intensity and the way in which depression affects older people will depend on many factors, such as the individual’s cultural background [6] and their social network [7,8].

In addition to this, according to some studies, 30% of people over 65 can suffer one fall per year and one in three of the elderly who suffer falls develop fear of falling again, which increases the chances that this will happen [9]. Older adults who live alone are more likely to suffer a fall due to having to carry out a greater number of activities at home and outdoors, compared to older people who do not live alone [10]. This is a very serious problem if, because of this, the person cannot get up, since they will likely go unnoticed by others. Another important factor to consider is that older people in one-person homes can suffer accidents or sudden consequences due to health problems that lead to the inability to move or even death, all without their relatives or neighbors being aware. A study on this topic was carried out in 1996 [11], and, currently, this continues to occur in some countries [12].

However, despite being aware of these risks, many older adults have voluntarily chosen to live alone to continue an independent lifestyle in their own home (especially in developed countries), while it is true that other older people simply have no other choice (either due to the lack of close relatives or for other reasons).

In relation to this, the Internet of Things (IoT) is a feasible solution for providing continuous and objective monitoring of the dependent person. There is currently no single definition for the Internet of Things (IoT), but it can be defined generally as a confluence of diverse technologies that provide Internet-based services and applications using electronic devices connected to physical items in order to collect data through sensors and control processes [13]. IoT technology is being used more frequently in healthcare, especially in the area of analysis of physiological parameters, observing investigations of human activity recognition, monitoring, anomaly detection, fall detection, sending alerts, etc. [14].

The present study describes the development of an IoT system called the SecurHome system, which allows older people to be assisted, while respecting their independence and privacy. One of the elements that integrates this system is the device for the home; this device is capable of detecting a data set using a wireless sensor network (WSN). It then sends this information directly to our server via the Internet (using the https protocol), to be stored in our database and subsequently processed to detect the pattern of user behavior and the pattern of environmental behavior in the user’s home. This allows a set of unusual rules of conduct to be created that the expert system will use to detect possible anomalies and alert family members and/or the nearest healthcare center by e-mail. Part of the set of rules to be used was designed by us, while there is another set of rules that, although they have a base structure designed by us, the expert who will create them will be the user (indirectly) through their activity over several days. Remembering that rule-based systems aim to encode the knowledge of an expert human, and enabling this through a set of assertions and a set of rules (IF-THEN), the system understands how it should act for a set of data obtained by the user [15].

Studies that have been conducted so far on the development of IoT assistive devices or systems focused on healthcare for the elderly have focused, variously, on wearable devices, biometric sensors, smartphones, applications, smart homes, environmental uses, indoor positioning, microphones, wearable cameras, and cameras [16]. In addition to hardware, regarding data analysis, studies may focus on low-level sensory data analysis (activity recognition and/or activity discovery); higher-level activity data analysis (anomaly detection and/or prompting system); or adaptation, trending, and concept drift [17]. For anomaly detection, different algorithms have also been used for data analysis [17]. In this paper, we have only highlighted some of the areas of research related to healthcare for the elderly, including only those that focus on their daily lives.

### 1.1. Fall Detection

The studies that focus on this area can also be subdivided, with some of them focusing on the development of wearable systems or devices [18,19,20], while others use edge devices (most commonly mobile phones) for information processing [21,22,23]. These products are usually highly effective in detecting falls. The problem with this type of device is that they can cause discomfort to the user, and some users may even consider them intrusive; in addition, they require the elderly person to remember to use them daily, which can be difficult if they suffer from some type of dementia. There are also studies focusing on the use of mobile phones and applications to collect data provided by the device’s accelerometer and gyroscope [24,25]. Other devices developed require the elderly person to press a button or take some specific action to trigger an emergency alarm, which is not feasible if the person is unconscious or is immobilized.

Other studies focus on non-wearable systems, most commonly by detecting falls through image or video processing [26,27]; many of them use deep learning methods [28,29,30,31], while others apply simple histogram and correlation methods, using networks of wireless video sensors to compare different frames to detect and confirm if the person has fallen [32]. However, elderly people may feel that their privacy is invaded, especially if this involves the installation of one or more cameras in their home.

The SecurHome system is not intended to detect falls as such. A fall, however, may be reflected in a change in the person’s habitual behavior, which will allow the system to send an alert to a family member, caregiver, or health center indicated for the care of the user. This has not been verified in this study, but we believe that it should be in future tests of this system, namely considering the possible addition of a new module to non-intrusively perform this task. Our system focuses on the two points described below.

### 1.2. Behavioral Monitoring

Among the literature found in this area, we observed studies focused on monitoring daily routines, behavioral pattern detection, and anomaly detection, using motion sensor networks [33,34]. Using binary sensors that only measure one type of information (motion) can be a limitation for the study of anomalies in the user’s behavior and in the home environment. For this reason, our home device includes binary sensors that detect different types of events (motion, light intensity, infrared to detect use of a remote control, fire, gas, and shock) and includes sensors that collect information periodically (temperature, humidity, pressure, butane and smoke levels). In addition to these studies, many other studies are focused on WSN systems, in which ambient sensors are placed on one or more objects and in rooms around the house, in order to record the interaction that the older person has with these objects throughout the day (pressure-based bed and/or chair sensors, contact-based door sensors, proximity-based toilet sensors, motion sensors, etc.) [35,36,37,38,39,40,41,42,43,44]; however, this option generally requires a significant installation of sensors in different points around the house, which can cause insecurities in the elderly, making them feel uncomfortable or invaded in their own home. For this reason, the device for the home that we have developed incorporates a wireless sensor network in a single device, which is intended to be placed in the room most used by the user. There are also studies that include body sensors and/or biosensors in their systems, which can enrich the data obtained about the elderly person, helping to prevent possible diseases and anomalies as a result of the person’s state of health [37,45,46,47,48]. However, for a monitoring system in the daily life of an elderly person, these biosensors must be additional to the proposed system or device, since they are usually devices that the user must wear, which implies that if the user does not wear it, it will not be able to collect information, and elderly people may forget to wear it or simply may not want to wear it.

There are also commercial devices that aim to take care of the elderly through home monitoring, such as GrandCare [49], which is a complete device, although it is not fully centralized, since the sensors must be placed on different objects in the home of the elderly person, which some people may consider intrusive. On the other hand, there is also AbiBird [50], which is much less intrusive but only uses motion sensors.

The device for the home that we have developed not only uses ambient sensors that measure periodically, but also sensors that are triggered by events. In addition, the data analysis includes not only the user’s activity, but also the user’s inactivity, information that, in line with the authors of previous works, we consider of great importance [44]. Regarding the sensors, these are non-intrusive (no video or audio recording) and involve low-power consumption. They are all concentrated in a single device, so a complicated installation throughout the user’s home is not required. This will all be explained in detail later. Using the device that we propose does not guarantee immediate detection of some types of emergencies, such as possible falls, illnesses, etc. However, these events may cause anomalies that could be detected, and being a completely safe device with respect to the user’s privacy and independence, it may find greater acceptance among these users, thereby increasing their health safety at home. To fulfill its purpose, our device should be placed in the room most used by the elderly person—the living room is most recommended. The installation is very simple as it simply needs to be connected to the power outlet and the Wi-Fi configured.

### 1.3. Reminders

Some studies investigate the best way to provide reminders or advice that will benefit the older person—for example, advising them to drink water if the temperature increases [51], to take their medication [52], etc. Other studies carry out this notification process by using information stored in a database (medications and/or medical appointments) [53,54].

Our device provides reminders about the medicine that the elderly person should take and the medical appointments scheduled for the day. This information can be read on the screen or heard through the device’s speaker so that the information reaches the person regardless of whether he or she has any visual or hearing difficulties.

This article is based solely on the design and development of the system. The next research step that we will carry out will, therefore, focus on the validation and improvements to be made to the SecurHome system hardware and software. To the best of our knowledge, this project is the first development of a system using a single device with a wireless sensor network of low-cost, non-intrusive sensors to obtain a variety of information regarding the most used room in the elderly person’s home.

In 2010, the European Commission advised European Union (EU) research and innovation projects to adopt the TRL scale: “The EU should apply the TRL scale R&D definition. (…) The Commission should also systematically apply this definition in order to include technological research, product development and demonstration activities within its RDI portfolio” [55]. Currently, our project is at TRL 6 stage—technology demonstrated in relevant environment (no real) [56,57]. Testing in a relevant environment was done in the home of one of the researchers due to restrictions relating to the ongoing COVID-19 pandemic. Therefore, we wished to involve as few external people as possible (mainly elderly people). The tests to be performed in the next stages of the study will be conducted in real environments (elderly people’s homes).

## 2. Materials and Methods

### 2.1. Materials

To describe in detail the materials used for the development of this system, it is necessary to have an overview of its structure. The developed IoT-based system is made up of a user assistance device that will be placed in the homes of users (elderly people who live alone); a web interface where caregivers and users can register and configure medication routines and medical appointments; and, finally, a server that handles and processes the information stored in the database. These three system components communicate with each other via the Internet. Figure 2 shows the general structure of the IoT SecurHome system.

### 2.2. Assistive Device for the Home

This is a device developed to be placed in the home of the elderly person. It will have direct contact with the user and is responsible for collecting environmental and behavioral data from the user’s home, all through incorporated sensors. It also allows interaction with the user by means of a touch screen, since the device has a graphical interface that allows multiple tasks to be performed, such as providing reminders about medication and medical appointment routines previously scheduled on the web (this information is stored in the database and the device for the home queries it daily). In addition, the user can send alert messages to their caregiver manually by simply pressing a button on the screen. It is made up of a control device (Raspberry Pi 3 B+) that is responsible for controlling all the instruments (sensors, speaker, buttons, and any other elements to be included in future updates) that are required to be installed in the device for the home. The control device handles two types of environmental sensors: those that collect information periodically, such as temperature and light intensity; and those that are triggered when they detect an event, such as motion, gas, or fire. In addition, it is responsible for managing possible interactions that the user will have with the device—for example, notifications about medical appointments and medications; the weather application; or Wi-Fi settings. It is also responsible for processing the values of the periodic measurements made by the sensors to be subsequently sent to the database, just as it directly sends the measurements of sensors that are triggered by events. Lastly, but in relation to the previous point, this allows Internet communication with the server for the storage and processing of data.

The hardware of the developed device is based on the use of a Raspberry Pi 3 B +, because it is a small, low-consumption device that allows us to efficiently control all the elements that we want to use. We installed the Debian-based open source Raspbian operating system (OS) [58]. In our project, the Raspberry Pi connects to the internet via Wi-Fi (2.4 GHz/5 GHz IEEE).

The device for the home also features a 7-inch touch screen, eight different sensors that will perform the desired measurements (these will be described in detail in the Methods section), a button to turn the device on and off, and a speaker to achieve better accessibility to the information supplied to the user. In addition to this, some other components such as resistors, an analog-digital converter, and a voltage converter from 5v to 3v are necessary, since the maximum input voltage allowed by the Raspberry Pi is 3.3V DC and some of the selected sensors have maximum output values of 5V or analog outputs.

The ports used in the Raspberry Pi were a USB port for the speaker and general GPIO ports for four of the eight sensors, while another of the sensors uses the Raspberry Pi’s I2C protocol and the last three require conversions since their output is analog and/or over 3.3 V. Figure 3 shows the device for the home without its casing, with the Raspberry Pi and the connected sensors all visible.

### 2.3. Web Interface

Users and their caregivers must be registered through a web interface; in addition, medication routines and scheduled medical appointments must be specified. This information is automatically sent via the Internet to be stored in the database.

### 2.4. Data Storage and Processing

This module is responsible for the storing and processing of all user data, has the necessary services to extract and store the information from the database, and also performs the task of creating rules for the operation of the expert system, taking into account the information detected by users throughout the day (this process is explained in more detail in Section 2.5). In addition, it stores the new rules or updates the modified rules in the database and is responsible for comparing the new information obtained by the home assistance device with the existing rules to detect if it is necessary to send an alert to the relative or caregiver. Similarly, this alert information is sent to the home assistance device so that it displays a message on the screen.

The expert system will have three general stages. The first is the calibration time, which lasts two weeks. Then, the data analysis and comparison with the existing rules are performed, in order to make modifications to the rules if changes are found in the pattern it had previously detected (in ranges of two weeks, over a month). After this, the study period is extended and the history of alerts generated is analyzed, to check if any alert is repeated with the same pattern; in the event of this happening, the rule that triggers the alert is modified.

### 2.5. Methods

For the creation of the SecurHome system, it was necessary to divide the research, design, and development work into the three essential aspects of the system:Data collection by means of sensors;User interaction and scheduled notifications about medicines and medical appointments;Creation/management of rules and sending alerts to family members or caregivers;Control test in the laboratory, to check the operation of the system.

### 2.6. Data Collection

First, it was necessary to carry out an investigation on the sensors to be included for the detection of anomalies in the home environment, in order to cover the greatest number of home emergencies at a low monetary cost, ruling out all those that could be considered intrusive by the elderly person. We have considered low-cost sensors that are not intrusive and that can provide information about possible accidents or behavioral anomalies in the home (movements at unusual hours, gas leaks or possible fire, use of heating at bedtime, among others). The analysis was carried out in two ways: first by considering the possible events that may occur in the home and looking at the sensors that could help us to detect them; then, we looked for other low-cost sensors that had not been considered, and we analyzed their possible usefulness for anomaly detection. Previous studies have detailed the environmental sensors commonly used for the care of the elderly [40]. We have included those that can provide relevant information while all being centralized in a single device in the home.

The sensors that we currently use are described in Table 1 (the possibility of including more sensors in future updates of the device is not ruled out, but currently, the device has one sensor of each type).

As shown in the table, there are two types of sensors used in the system: those that perform periodic measurements (every minute) and those that are triggered when an anomaly is detected. This information is collected and sent directly to a server that stores it in the database. The shock sensor, similarly to the other sensors included, is integrated into the device; with this sensor, we will be able to know if the device has received a shock (either an accidental shock, because it has been dropped, or any other cause).

Most sensors are connected to the Raspberry’s generic input GPIO pins, except for the temperature and atmospheric pressure sensor, which uses the Raspberry’s I2C protocol (connected to the SCL and SDA pins), and analog sensors, which are connected to the analog–digital converter. This stage of the project was developed in Python 2.7.

### 2.7. User Interaction and Notifications

In the first interaction with the user, it will be necessary to configure Internet access; a drop-down menu will appear, where the Wi-Fi network of the home can be selected. Then, a keyboard will appear, which the user must use to enter the password. At the end of this process, by default, the device will display a screen with the weather in the user’s area. These features can be seen in Figure 4.

If the user does not want the device to make audible notifications (through the speaker), it is possible to turn off the speaker by clicking on the sound icon seen in Figure 4b. As mentioned above, the device offers two types of notifications, indicating to the elderly person when they should take their medication (name of the medication and amount to take) and also indicating when they have a medical appointment. The latter reminder is given twice, the day before the appointment (24 h before) and three hours before; this one specifies the type of medical appointment, and the time and location. All this information must be previously stored in our database and will have to be entered by the user themselves or by a caregiver, family member, or doctor (depending on the user’s level of dependency) through a web interface, which is currently under development. Figure 5 shows the two types of notifications that are provided by the device for the home.

It should be remembered that this information is also supplied through the speaker, so if the user is not in the same room as the device or if they are otherwise visually impaired, they can still receive the notification without any difficulty. The speaker connects to the Raspberry via a USB port. This stage of the project, similar to the previous one, was developed in Python 2.7.

### 2.8. Rules and Sending Alerts

The creation of rules for the expert system is done with the information being collected from the users in a period that we call “calibration”; after this, they enter a period of readjustments and finally the stable period. The system has two types of rules: permanent rules (if there is the presence of gas in the home, send an alert message) and modifiable rules, which will vary during the daily life of the older person until they achieve the pattern that most closely resembles that daily life. The modifiable rules have a structure that is already formed (defined with default values) that covers the behavior of a full day; these will be modified, and these changes will depend not only on the data obtained but also on the hours in which they were obtained.

The rules defined by default were chosen after an initial data collection in the home of one of the researchers, which was of two weeks’ duration. With this, we only plan to obtain a set of structured rules; the values set to these rules are not of great importance, since they will be modified in the future by the habitual behavior of the elderly person. Figure 6 shows the behavior of the data collected by two of the sensors installed in the device (movement and interaction with the TV remote control).

This is an example of a couple of measurements collected, with the behavior presented in Figure 6. We set up four rules for movement (two structures, presence of movement in an hour range, and non-presence of movement in an hour range) and two for interaction with the TV remote control (single structure, interaction with the TV remote control in an hour range):Send an alert message if there is movement during unusual hours (00:00–8:00);Send an alert message if there is no movement during usual hours (9:00–12:00);Send an alert message if there is no movement during usual hours (16:00–18:00);Send an alert message if there is movement during unusual hours (23:00–00:00);Send an alert message if there is interaction with a remote control during unusual hours (00:00–8:00);Send an alert message if there is interaction with a remote control during unusual hours (22:00–00:00).

In the calibration period, the device must monitor the elderly person for two weeks without sending any type of alert message to the relatives or caregiver. This will generate the first 14 samples (one sample per day over two weeks), which will be processed to generate the first behavior pattern before starting to send alerts. Throughout the first day of calibration, information will be collected that will be used to create the monitoring rules for the first time. Every time a calibration day ends, the newly generated rules will be compared with the previous ones, expanding or decreasing the hour ranges of the previously created rules or creating new hour ranges for each combination of previously structured measurements. The readjustment period has the same operating principle as the calibration period, in which the set of rules will continue to be changed considering the measurements collected throughout the day. In this way, the first two weeks of samples will be considered, and the process will continue for an additional two weeks; however, the system can already send alert messages if it deems it appropriate throughout the day.

Finally, in the stable period, it is still possible to make rule changes; however, any change must be studied in depth before being made. When a behavior change occurs, a message will be sent and that behavior will be recorded in case it is repeated in subsequent days; this is done in order to maintain the training of the set of rules. If a certain type of alert is constantly repeated in the same pattern (day of the week or time of the day), the behavior is evaluated, and a modification of the rule is made. Table 2 shows in detail the structure of the designed rules. 

These are not rules of habitual behavior, but rather of unusual behavior; however, they are created using data obtained from the habitual life of the elderly person. It is also necessary to highlight that there are two types of messages that can be sent to the person or entity responsible for the elderly person: these are alerts and messages about a possible emergency.

In addition to the rules described above, a set of meta-rules have also been designed that have priority when sending alert messages. These meta-rules check the behavior of two measures; if both are fulfilled in the same period, a single notification is sent to the caregiver or family member of the dependent. Table 3 shows some of the defined meta-rules. Similar to the rules described above, these meta-rules can be modified depending on the usual behavior of the elderly person.

### 2.9. Control Test

Control tests of each of the stages of the system have been carried out with a test user, checking the complete flow of the SecurHome system. The configuration of the sensors, the constant connection to the server via the Internet, and the correct periodic storage of the values detected in our database for a test user were checked, after which the medicine notification process was also checked, as was the notification of medical appointments. To do this, it was necessary to previously configure a medication routine and some medical appointments to our test user, so that the information was in the database. Finally, in the control tests, we also checked the system’s operation by detecting unusual behavior considering the rules defined by default, and we verified the process of sending alerts to the indicated caregiver. In this test, we considered that the default rules are the definitive rules. One of the researchers was monitored by the device in her home environment over a period of 2 weeks without intentionally altering the environmental conditions of the room; subsequently, on a single day of study, the researcher altered the environmental conditions of the room, to test the behavior of the system in extreme cases not presented in the previous two weeks. Finally, we evaluated the response of the system by obtaining the different behavioral signals in both periods studied.

## 3. Results

This section details the results obtained in the control test carried out throughout the study period. The purpose of this control test is to verify the operation of the device and the system in detecting unusual behavior and sending alert messages, considering the rules of unusual user behavior. It should be remembered that these predefined rules in the system are totally modifiable, with the purpose of adapting them to the daily life of the dependent person. However, for this system test (performed in a controlled environment), the predetermined rules and meta-rules were used as the definitive rules.

### 3.1. Measurement of Movement and Interaction with the TV Control

We analyzed the behavior of the data collected on the motion detected in the control user’s home during the two weeks of study without intentional alterations in the room environment, considering the rules that were defined by default as the definitive rules for this control test (the same set of rules as described in the Rules and Sending Alerts section):Send an alert message if there is movement during unusual hours (00:00–8:00);Send an alert message if there is no movement during usual hours (9:00–12:00);Send an alert message if there is no movement during usual hours (16:00–18:00);Send an alert message if there is movement during unusual hours (23:00–00:00).

Regarding the interaction with the TV remote control, the same set of rules described in the Rules and Sending Alerts section were considered (also modifiable according to the user):Send an alert message if there is interaction with a remote control during unusual hours (00:00–8:00);Send an alert message if there is interaction with a remote control during unusual hours (22:00–00:00).

Figure 7 shows the motion detection (dark blue) and interaction with the TV remote control (light blue) over two weeks (starting on Wednesday), for 24 h each day. In addition, Figure 7 shows the alert messages that were sent by the SecurHome system (orange).

In the two weeks under study, the system sent ten alert messages when witnessing motion in the 00:00–08:00 time range; in addition, it sent one alert message when not witnessing motion in the 09:00–12:00 time range, it sent four alert messages when not witnessing motion in the 16:00–18:00 time range, and it sent seven alert messages when witnessing motion after 23:00. The system also sent two notifications as the test user interacted with the TV remote control at 22:00 on two days during the period under study. The two no-motion alerts on the second Saturday of the study occurred because the person under study left home in the evening of the previous day and did not return until Saturday afternoon. These alerts are useful for reporting that the elderly person has not returned home or has alternatively passed out in their own home. On the first Thursday, there were several motion alerts messages in the early morning; this occurred because the user had stayed awake in the living room with her laptop, and on the first Monday, the user could not sleep in her bed and lay down on the sofa in the living room for a change of environment. It is important for these last two behaviors to be reported, especially if the user is an elderly person living alone.

Regarding rule modifications, in this pilot test, the system did not make modifications, but in this result, we observe behaviors that should be evaluated by the system for possible modifications of the rules. For example, on the two test Sundays, alerts were sent for an absence of movement, as the person under study was out of his home all afternoon. If this behavior is repeated for another two weeks, the system should modify the rule for Sundays. The system should also modify the range of hours for TV viewing, since these two weeks were not as active as the calibration week, and possibly evaluate the range of movement at night, as it can increase the range until 00:00 (considering the number of alerts sent for movement after 23:00). The time it takes for the system to send a second message of the same alert or notification can be configured, currently, and by default, it is 30 min.

### 3.2. Measurements of Temperature

The temperature analysis was carried out throughout the two weeks under study; there was no intentional alteration in the environment of the room where the device for the home was located. The rules that were considered for sending alerts for sudden changes in temperature are described below.

Send an alert message if the temperature exceeds 38 °C;Send an alert message if the temperature is less than 15 °C;Send an alert message if the temperature rises more than 2 °C between two consecutive measurements;Send an alert message if the temperature at bedtime (23:00–7:00) is higher than or equal to the highest temperature during the day (9:00–16:00);Send an alert message if the temperature rises by 1 °C between two consecutive measurements, and there is also fire in the room.

Figure 8 shows the behavior of the temperature during the two weeks studied. It also shows the alert messages sent considering the rules described above; these messages are represented in binary form, since they are sent by the triggering of an event. The SecurHome system sent three alert messages; two of these were on the same day (early morning of the first Thursday) and the third was on the second Tuesday of study. All three alert messages were sent due to the detection, at the user’s usual bedtime, of temperatures higher than the maximum temperature detected during the past afternoon. On the afternoon of the first Wednesday, a maximum temperature of 21.8 °C was detected; then, on the first Thursday, the system detected temperatures between 21.8 °C and 22.0 °C several times between 2:01 and 3:33, so it sent two alert messages. In addition, on the afternoon of the second Tuesday studied, a maximum temperature of 23.8 °C was detected; then, the device detected temperatures between 23.8 °C and 23.9 °C between 23:17 and 23:32 on the same Tuesday and sent an alert message.

On the first Thursday, as observed in the motion result, the user stayed awake in the living room in the early morning, so she left the heating on during that time. On the second Tuesday, the user went to sleep and left the heating on while sleeping. The latter case can be dangerous, so the alert is extremely useful.

### 3.3. Sending Alert Messages with Intentional Alteration of the Environment

For the following experiment, three types of alteration were made to the environment of the room where the device for the home was placed. The temperature was intentionally and abruptly altered, and the device was placed in the presence of fire and gas on several occasions to observe the behavior of the system. Figure 9 shows the behavior of the temperature throughout the day of the experiment and the alert messages sent by the system considering only the changes in temperature. The alterations in the environment began to take place around 12:00 on the day of the experiment. Sudden temperature increases occurred in the following time ranges: between 13:51 and 14:00; between 19:29 and 19:36; and between 21:54 and 21:59. The caregiver received alert messages for this unusual behavior at 14:00, 19:36, and 21:59, respectively. The system currently sends the alert messages almost immediately; this will probably be modified in future work, to balance the server workload and the response time for sending alert messages.

In addition to these results, Table 4 describes the data obtained for the presence of fire and the presence of gas and shows the alert messages sent for fire and gas events, respectively. Regarding the fire alert messages, these are not sent only for the presence of this event, since, when the system perceives fire, it performs an analysis of the temperature to check if it has increased. If the temperature rises and fire is present in the room, a fire alert message is sent. Sending alert messages only for the presence of fire in the room has not been considered, due to the possible false positives obtained by this sensor.

Regarding the alert messages sent for the presence of gas, the system does not consider any other measure; if the device senses gas, the system sends an alert message to the user’s caregiver. The time ranges in Table 4 have been divided into 30-min ranges, since, in theory, alert messages for the same type of event are not sent twice in a row until 30 min have elapsed. Additionally, the ranges of hours during which there was no presence of fire or gas have been eliminated from the table.

On this day of additional experiments, there were no false positives from the fire sensor; the false values may have occurred due to natural room lighting or other factors, which will have to be studied and solved in future work. However, as shown in Table 4, only one alert message was sent for the presence of fire; this occurred because the presence of fire (21:56) coincided with the temperature increase that occurred between 21:54 and 21:59. The system sent an alert message for the presence of fire at 21:57. No other fire alert messages were sent because there was no presence of fire in the other two temperature increase ranges. Between 13:30 and 14:00, fire was present at 13:40 and 13:39; however, the temperature increase occurred between 13:51 and 14:00. Regarding the range of hours between 19:00 and 20:00, there was no presence of fire.

Moreover, as expected, only one alert message was sent per 30-minute range (even when the device detected gas in the room consecutively or more than once in those hour ranges). Table 4 shows that in the range between 14:00 and 14:30, there were no alert messages, even when gas was present; this occurred because the gas event was detected at 14:29. In the previous range, however, an alert message for the presence of gas had already been sent, at 13:56, so 30 min had not yet passed for another alert message to be sent. The same occurred in the 21:00 to 21:30 time range.

## 4. Conclusions and Future Work

An anomaly-monitoring, detection, and alert system based on the installation of the Internet of Things (IoT) has been successfully implemented. This system is based on the use of a Raspberry Pi and, through the developed software, is capable of detecting information through various sensors, storing the information detected through secure communication with the server, and then processing the stored information to generate and/or update rules that allow possible unusual behaviors throughout the day to be detected, as well as sending alert messages when these behaviors are detected.

This is an unobtrusive, easy-to-install, and low-cost system that involves the greatest number of measurements to date to be considered for the detection of a dependent person’s behavioral patterns, involving not only periodic measurements by environmental sensors, but also event-driven measurements. This project has been a first step in the development and installation of low-cost, non-intrusive IoT systems in the homes of older people, to detect anomalies in their daily behaviors, in addition to providing notifications that will allow them to have better control of their health.

This project will continue to perform control tests with the data collected over a longer period, until a robust prototype is obtained in a relatively controlled environment. The next step in the project will be to validate the device with real data. This validation must be carried out with two analyses: a first analysis to validate the operation of the system (creation and modification of the rules adjusted to the person’s daily life, and generation of the different notifications), for which it will be necessary to carry out the installation of the system in at least one single-person dwelling (preferably in which a person over 65 lives); and a second analysis on the acceptance of the system by the target audience. For this, a qualitative study must be carried out through surveys or other means; it would also be interesting to evaluate the usability of the developed device and assess proposals for improvements.

For future work, it will be necessary to consider some potential problems. The presence of visitors in the home could alter the user’s behavior pattern, and the system could even send alert messages for unusual behavior, presenting data that are not representative for the purpose of this project. The presence of visitors should be detected, either manually, where the user or their caregiver indicates that there are visitors in the home, or in some other way—this is an opportunity for future work.

For future upgrades, a mobile application (Android and iOS) is also being considered as an additional component of the SecurHome system. This will be useful in two areas: notifications (medication and medical appointments) can be received directly on the elderly person’s phone; and in the area of accident and anomaly detection, information provided by the phone’s internal sensors (gyroscope and accelerometer) can be included, which has been extensively studied in previous work for fall detection, since the device is not capable of detecting falls that a person may suffer.

In addition to this, a module will be developed to stop the alert messages considering the rules generated for the user in case the person is absent for a long time—for example, on vacation. However, some alerts will continue to be active—for example, the presence of gas and fire—but the environmental behavior in such a period will not modify the user’s behavior rules.

This study is also limited by the nature of the data and the method used for data analysis. The data used may present noise due to the amount of information and possible non-relevant data, which are difficult to handle with the currently used methodology of rules. The use of other data analysis algorithms for anomaly detection is proposed for future work.

The current study was limited in the amount of data used, due to the time in which this first control test was performed. In addition, during the period under study, there were some false-positive results generated by the fire sensor; these occurred as one-off events on six of the 15 days studied, and on the last day, there were no false positives. Thus, we must also consider solving the malfunctioning of some sensors, which can affect the system’s detection of the behavior patterns. This should be done automatically, filtering the information that is collected by detecting data that do not reflect the real behavior of the user or the home environment. After these problems have been solved, the system will continue to be tested in a controlled environment until a robust device is obtained for subsequent evaluation in real environments.

In addition, as this was a pilot test carried out in the home of one of the researchers, we could not observe all the real limitations and problems that the device may present in a real environment (the one-person home of an elderly person). However, this is a first step for this project, as updates are constantly being made and further testing of the SecurHome device and system will be necessary.

## Figures and Tables

**Figure 1 sensors-21-05454-f001:**
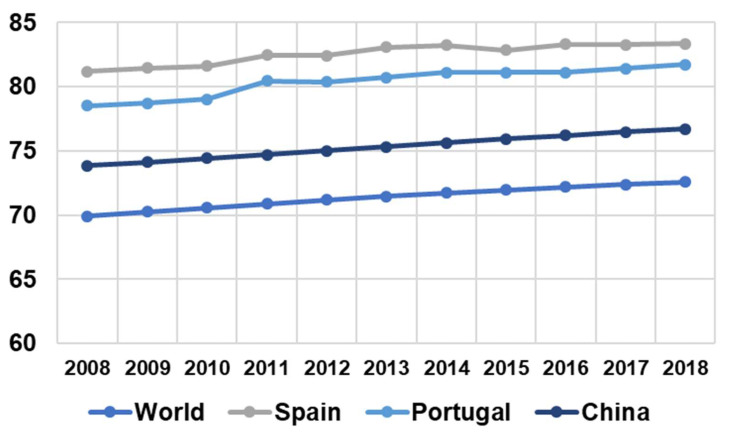
Statistics of the last 10 years on life expectancy at birth in the world and in some specific countries.

**Figure 2 sensors-21-05454-f002:**
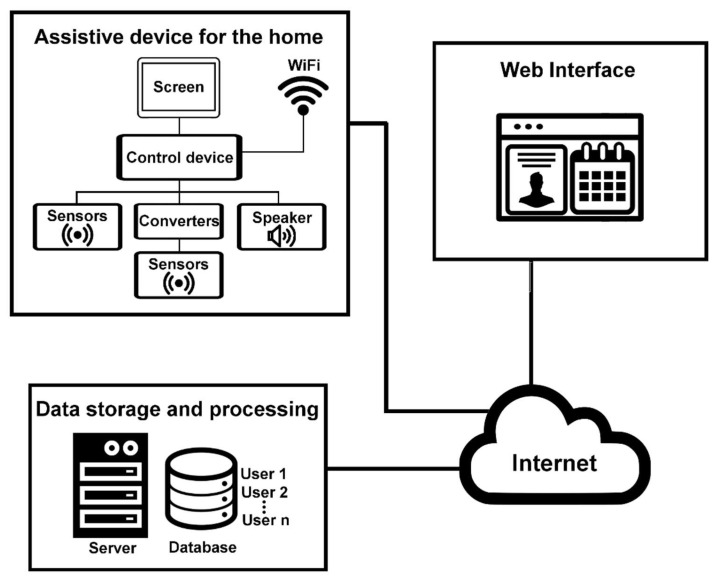
Structure of the IoT SecurHome system.

**Figure 3 sensors-21-05454-f003:**
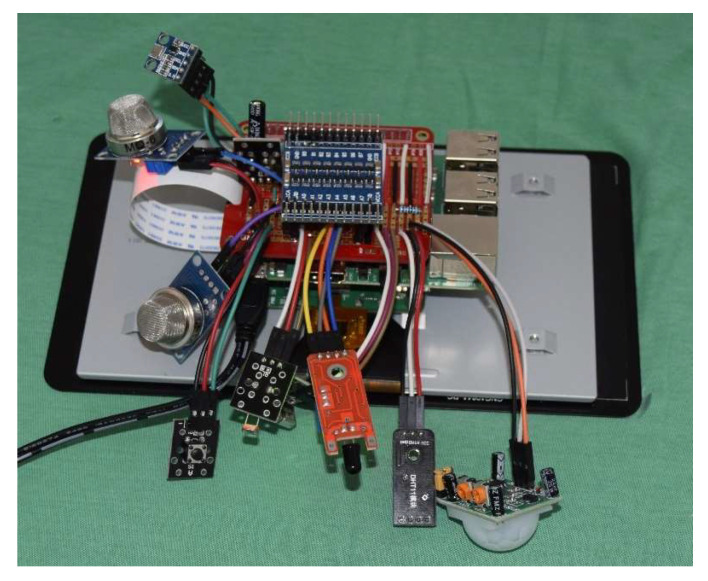
Connections of the elements that make up the home assistance device.

**Figure 4 sensors-21-05454-f004:**
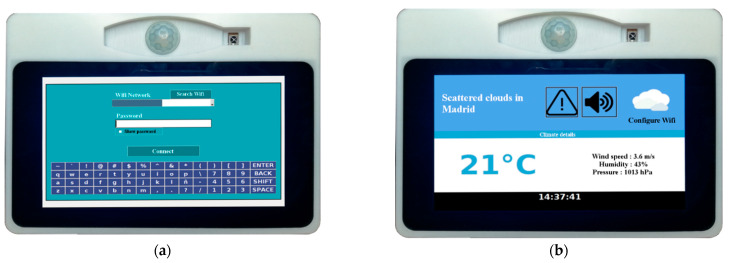
(**a**) Screen to configure the Wi-Fi network of the house; (**b**) Screen to check the current weather.

**Figure 5 sensors-21-05454-f005:**
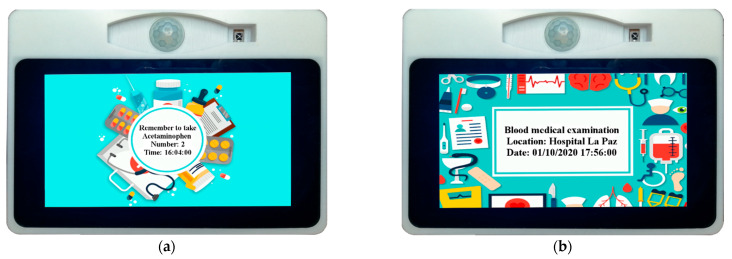
(**a**) Notification about taking medicine; (**b**) Medical appointment notification.

**Figure 6 sensors-21-05454-f006:**
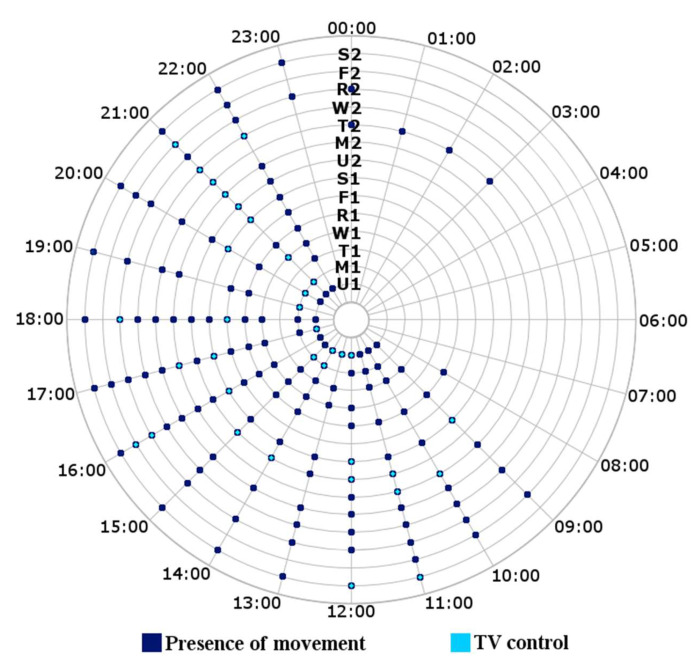
Movement behavior and interaction with the TV remote control to structure the default rules of the system. The axes show the time over the course of a day and the days of the week from the first Sunday (U1) to the second Saturday (S2) of calibration.

**Figure 7 sensors-21-05454-f007:**
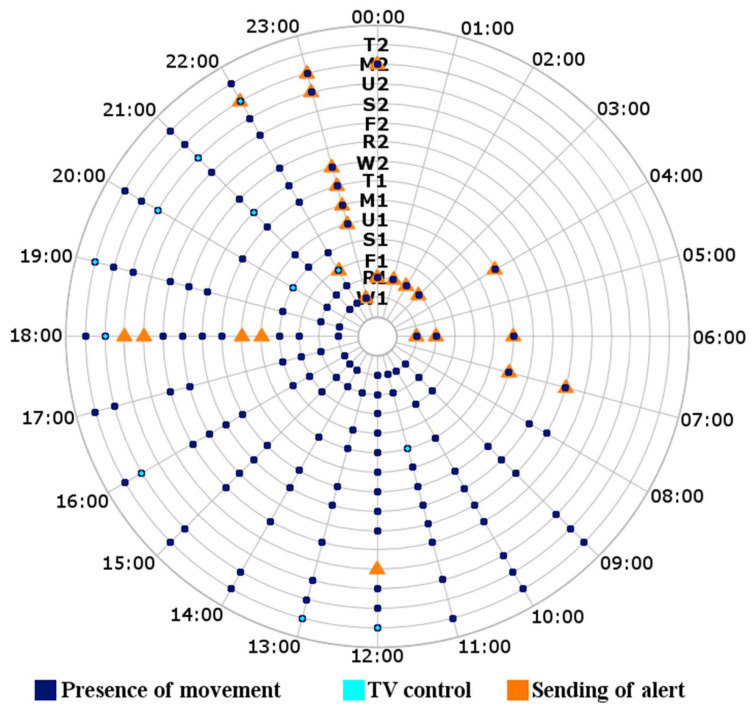
Movement behavior and interaction with the TV remote control in the control test. The axes show the time over one day, and the days of the week in the period studied from the first Wednesday (W1) to the second Tuesday (T2).

**Figure 8 sensors-21-05454-f008:**
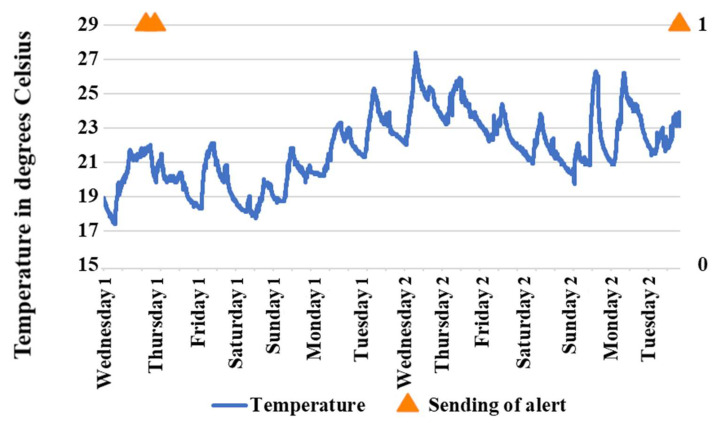
Temperature behavior in the two weeks studied in the control test.

**Figure 9 sensors-21-05454-f009:**
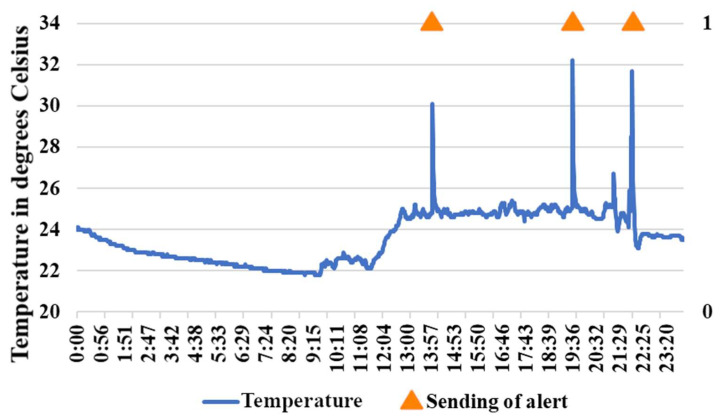
Temperature behavior and sending of alert messages in the intentional environmental alteration experiment.

**Table 1 sensors-21-05454-t001:** Sensors used in the device and their characteristics.

Sensor	Characteristics
Temperature and humidity sensor	Detects ambient temperature and humidity periodically
Barometric pressure and temperature sensor	Detects ambient temperature and atmospheric pressure periodically
Photo sensor	Periodically detects lighting level
Gas sensor (CH4, butane, propane)	Detects gas levels (CH4, butane, and propane) periodically and provides the binary state of gas presence in the coverage area (event)
Motion sensor (infrared)	Provides the binary state of motion presence in the coverage area (event)
Fire sensor	Provides the binary state of fire presence in the coverage area (event)
Infrared receiver sensor for TV remote control	Provides the binary state of infrared light emission from a remote control (event)
Shock sensor	Provides the binary state of presence of movement or direct hit to the device (event)

**Table 2 sensors-21-05454-t002:** Rules of unusual behavior to alert the family member or caregiver of the dependent person.

Rule	Type	Characteristics
Temperature < minimum value	Modifiable	By default, the minimum value is 15 °C, but this rule is modifiable depending on the normal behavior of the user. It is always checked that the temperature does not decrease from the set minimum value. If this happens, an alert message is sent.
Temperature > maximum value	Modifiable	By default, the maximum value is 38 °C, but this rule is modifiable depending on the normal behavior of the user. It is checked all the time that the temperature does not exceed the maximum value set. If this happens, an alert message is sent.
Gas presence	Permanent	The presence of gas is checked all the time; in case it occurs, an alert message is sent.
Presence of fire	Permanent	The presence of fire is checked all the time: in case it occurs, an alert message is sent.
Presence of movement in an unusual time range	Modifiable	If movement is detected in an unusual time range, an alert message is sent.
No presence of movement in the usual time range	Modifiable	If no motion is detected in a time range in which it should usually occur, an alert message is sent.
Temperature increases (modifiable value) in one minute	Modifiable	By default, the value is 2 °C. Compare the current temperature measurement with the temperature detected in the previous measurement; if it increases by 2 °C (modifiable), it sends an alert message.
Nighttime temperature is higher than daytime temperature	Modifiable	The system stores the highest temperature in a daily time range (9:00–16:00). If the current time is equal to or greater than the time at night when there is usually no user movement present, and the current temperature is higher than the stored temperature, an alert message is sent for heating on at bedtime.
Interaction with the TV remote control in an unusual time range	Modifiable	If the use of a TV remote control is detected within an unusual time range, an alert message is sent.
Light intensity < minimum value, in unusual time range	Modifiable	By default, the minimum value is 20% brightness for a nighttime range (18:00–21:00), but this rule is modifiable depending on the normal behavior of the user. It is checked within the time range under study that the light intensity does not decrease from the set minimum value. If this occurs, an alert message is sent.
Light intensity > maximum value, in unusual time range	Modifiable	By default, the maximum value is 1% brightness for a nighttime range (23:00–5:00), but this rule can be modified depending on the normal behavior of the user. It is checked within the time range under study that the light intensity does not exceed the maximum value set. If this occurs, an alert message is sent.

**Table 3 sensors-21-05454-t003:** Meta-rules of unusual behavior to alert of possible emergency.

Rule	Type	Characteristics
Presence of movement and presence of interaction with the TV control	Modifiable	If movement and interaction with the TV control is detected within the same unusual time range, an alert message is sent indicating that the person is watching TV at an unusual time.
Presence of movement and low light level	Modifiable	If motion is detected in a low-light environment within an unusual time range of movement, an alert message is sent.
Presence of fire and significant temperature increase	Permanent	If a fire is present and the temperature increases by a specified number of degrees within a time interval, an emergency message is sent.

**Table 4 sensors-21-05454-t004:** Alert messages with intentional alteration of the environment.

	Presence of Fire		Presence of Gas	
Range of Hours	Number of Triggered Events	Alert Message Sent	Number of Triggered Events	Alert Message Sent
11:30–12:00	1	-	0	-
12:00–12:30	0	-	3	1
12:30–13:00	9	-	0	-
13:00–13:30	6	-	4	1
13:30–14:00	2	-	4	1
14:00–14:30	1	-	2	-
14:30–15:00	1	-	2	1
15:30–16:00	0	-	2	1
17:00–17:30	0	-	3	1
18:30–19:00	0	-	2	1
20:30–21:00	0	-	2	1
21:00–21:30	0	-	2	-
21:30–22:00	3	1	2	1

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
