# Peer review of "Development of an Anomaly Alert System Triggered by Unusual Behaviors at Home"

_sensors, 2021, doi:10.3390/s21165454_

Round 1

Reviewer 1 Report

Overall, you present an interesting experiment around the use of the SecureHome system, an IoT based system for monitoring the activity of elderly people living alone at home.

While the work is presented clearly, you make no attempt to explain how your approach differs when compared with other similar projects. What is different, new, important about the work you are presenting?  I personally am not convinced that you offer anything new in this system - you certainly don't highlight the novelty.

The evaluation seems to have been carried out in the home of one of the researchers over a period of a number of weeks, with the first 2 weeks being a calibration period.

Overall, I find the results a little underwhelming - it is not clear what you are showing, other than that the SecureHome system "works" as expected.  The authors themselves point out a number of next steps, including: evaluation with real data, qualitative surveys of users, consideration of the impact of visitors on data. For me, these are the interesting outcomes of this project and the things I would like to see publications about.

In summary, you need to rewrite the paper to be clearer about the contributions your approach makes to the state of the art and the insights you have gained from the experiments you have carried out.

Reviewer 2 Report

The paper proposes a sensor-based system in a smart home with the aim of detect anomalies that may trigger some alerts to the family members. 

Introduction: 

  • Spanish data are described; similar data can be found for more extensive territories, e.g., for Europe. Even if Fig. 1 reports some data also of other countries, the text is too focussed on Spain. I suggest considering also European/World data.
  • Line 39: it is meaningless to know the increase of the population without knowing the population of the whole country; a percentage would be more meaningful. 
  • Line 49: what does it mean "and abroad"? Did you mean outdoor? 
  • Lines 86-105: here, the authors focus on the problem of fall detection, which is not directly considered in the proposed system (an alert is sent if there is a change in the person's usual behaviour, which may be due also to a fall); however, in the rest of the paper, no use case about falls detection is described. So, it is not clear, why this subsection and all the related work about this problem has been reported. 
  • Lines 106-117: the literature starts with the detection of activities of daily living. This part is very general, the authors could better highlight the possible gaps they found in other systems that can be improved with the proposed system.  
  • line 117-120: This part is not clear at all, because the authors just refer to a "device" (what is this device? A wearable device?) with "non-intrusive sensors" (which sensors?). From the introduction, it is not clear what the paper is going to propose and what this "device" will do. I suggest improving the introduction, by better explaining
    • the problems you try to solve with this proposal (fall detection? not from the rest of the paper)
    • what you propose (a device? Is this correct? A system? An algorithm?) 
    • Finally,  a list of innovative aspects of this paper should be highlighted, because to me the novelty of the proposed approach is not very clear. 

Materials and methods

  • Lines 146-147: at this point there is a reference to the medication routines and medical appointments: is this one of the aims of the system? How is it related to the sensor-based part? 
  • Lines 153-155: what is the "IoT device" that is made up of a "control device" that controls the "instruments that are required to be installed in the final device"? What is this "final device" and are these "instruments"? The architecture must be described in a more precise way. 
  • Line 158: does this "IoT device" provide "notification of medical appointments and medications"? How? What is this IoT device?
  • Lines 164-167: at this point, something becomes clearer, but up to this point, the reader cannot imagine what you are proposing. I suggest revising this section by explaining at the one side the system at a high level (in terms of functionalities) and on the other side the hardware
  • Line 189: an "expert system" is created by the authors. Usually, in an expert system, there is an inference engine that deduces new knowledge. From the paper, it seems that the authors simply propose a rule-based system, where rules are more similar to IFTTT rules than to the rules of an expert system. The authors should better clarify their approach (also in the next sections) and better explain how the system infers new know knowledge. Is it limited to some adjustments? Are these adjustments automatic? See also comments below.  
  • Line 214: the authors "carried out an investigation on the sensors to be included for the detection of anomalies in different environmental aspects of the home". Which are these "aspects"? Which are the most interesting anomalies for the target users? The temperature and humidity? According to which users requirements? The shock sensor? Where is it placed and at which aim? I would expect to have first some requirements of the system and then the choices that were made according to the requirements. Instead, I have the impression that given the typical sensors, some rules have been identified. The authors should provide a more extensive explanation of the system starting from the users' requirements or motivating the choices they did. 
  • Given this list of sensors, how did you organize the experiment? How many sensors of each type did you use? Just one sensor per type connected to a single Raspberry placed in the living room? Where did you position the sensors? 
  • What does it happen if the user does not see/hear the messages notified by the system? 
  • Figure 6: what are the letters in the column below time 00:00? 
  • What does it happen if habits change? For example, because there are holidays? Do you always consider a sliding window of two weeks? 
  • Line 270: Two weeks are needed for calibration: how did you decide this time? According to which experiments? Once you collect more data, do you still limit to 2 weeks or do you consider a more extensive period?
  • Lines 282-286: "in the stable period it is possible to make rule changes, however, that change must be studied in depth before making the rule change". Who changes the rules and how? 

Results:

  • Results do not include anything about falls, which seem an important problem at the beginning of the paper
  • For the behaviour of the user, the system seems to have some limitations. First of all, it was tested by a single user, for a limited amount of time (in his/her home? living alone? at home for the whole day? with the same habits in the same time every day?). It would be interesting to test it in a real setting to better understand the limitations of the proposed approach.  Moreover, the results section mainly reports a list of rules with some diagrams reporting the detected events (plus how many alerts have been sent), so it is clear that the system "works", but not if it was actually useful for the user. I do not see interesting discussions in the results part. For example, there were some false positives: why? Is it possible to correct them by means of rules? If a person falls, how much time is needed for the system to understand that there is an anomaly? The alerts that were sent, to which anomalies corresponded? Just a different behaviour compared to the typical behaviour? For example, usually, I watch tv at a given time but one day (e.g., on Sunday) I switch it on at a completely different time: is this detected as an anomaly? The same applied to motions/non-motions. How many of the alerts were useful to be known? 
  • Figure 7: you may use different symbols for alerts (not circles, for example, triangles?) to improve the readability of the figure.  

Discussion

  • The correct title for this section should be Conclusions and future work since it does not discuss anything of the proposed system. 
  • Which are the limits of the approach? There are a few lines (440-444), but I believe that the discussion could be extended. 

References

  • There are many references on falls detection: did you try to reproduce it to get an anomaly? 
  • I was quite surprised that some researches have not been cited at all (e.g., Diane J. Cook did a lot of research on such topics) and that in the paper no commercial system has been cited (nowadays, there are different proposals of systems that detect anomalies - e.g., Canary Care, but there are many others)

For the references I suggest considering: 

  • The works by Diane J. Cook, which include also works on anomaly detection
  • For the anomalies, I believe that it may be of interested this work https://dl.acm.org/doi/10.1145/3342428.3342658 (which discusses also the case with few sensors) and surveys like https://link.springer.com/chapter/10.1007/978-3-319-21671-3_9

English: an English revision is needed.

Round 2

Reviewer 1 Report

I am broadly happy with the changes made.  I found 1 minor typo:

Line 169 has a typo: " (…) Th"

Reviewer 2 Report

The new version of the paper has been improved. I still believe that results could be improved, but in general the paper can be accepted. 

This manuscript is a resubmission of an earlier submission. The following is a list of the peer review reports and author responses from that submission.

Round 1

Reviewer 1 Report

  1. Figures 4 and 5 require an explanation or translation in English.
  2.  In figure 6 the labels for the days do not make sense. They require some  clarification. 
  3.  A test of two weeks by a single person is not enough to test the robustness of the system under  real user's  conditions. A plan or recommendation  for more comprehensive testing should be included in the future work comments.

Reviewer 2 Report

Dear authors,

I would highly recommend you to get more experience in writing a scientific manuscript as well as state of art in the smart home!

  • In throughput of the manuscript, you have misused invasive and non-invasive. What you mean is unobtrusive!
  • basically, a simple wearable device can detect a fall which is non-invasive and this aspect is not relevant, at all!
  • you have used "on the other hand", several times! this is used for a comparison! please use terms, terminology, and technology in the right way!
  • the sensors you have used are the most basic which do not have any contribution to detecting emergencies!!! Do you believe vital signs monitoring is the most significant?
  • if you would have used at least Raspberry pi 4, which supports both 3.3 and 5v, you wouldn't need a DC-DC converter!
  • Describing the basic terms (IIC, ADC,...) in a technical journal such as sensors do not have any contribution.
  • You have missed the most important papers in the field! I suggest carefully review the literature!

I regret but the manuscript is not scientific nor suitable for the community but rather a tutorial for high school students!

best regards and good luck